# Immunogenetic Predisposition to SARS-CoV-2 Infection

**DOI:** 10.3390/biology12010037

**Published:** 2022-12-25

**Authors:** Claudia Lehmann, Henry Loeffler-Wirth, Vera Balz, Juergen Enczmann, Ramona Landgraf, Nicole Lakowa, Thomas Gruenewald, Johannes C. Fischer, Ilias Doxiadis

**Affiliations:** 1Laboratory for Transplantation Immunology, University Hospital Leipzig, Johannisallee 32, 04103 Leipzig, Germany; 2Interdisciplinary Centre for Bioinformatics, IZBI, Leipzig University, Haertelstr. 16–18, 04107 Leipzig, Germany; 3HLA Laboratory, Institute for Transplantation Diagnostics and Cell Therapeutics, ITZ, University Hospital Duesseldorf, Moorenstr. 5, 40225 Duesseldorf, Germany; 4Clinic for Infectious Diseases and Tropical Medicine, Chemnitz, Flemmingstraße 2, 09116 Chemnitz, Germany

**Keywords:** COVID-19, classical HLA, non-classical HLA, MHC, HPA, blood groups, SARS-CoV-2, immunogenetics

## Abstract

**Simple Summary:**

Since the beginning of the SARS-CoV-2 pandemic in 2020, numerous data with respect to the influence of immunogenetics on the predisposition to and severity of infection have been reported worldwide (PubMed; *n* = 228; 6 November 2022). Immunogenetics play a pivotal role in infection, vaccination, its failures, and/or vaccination breakthrough. Factors including the major histocompatibility complex and the common ABO blood group system have been discussed. Herein, we describe the association of HLA-A, B, C, DRB1, DRB345, DQA1, DQB1, DPA1, DPB1, and HLA-E, F, G, and H on the results of molecular detection of COVID-19, or, in some cases, on antibody detection upon first testing. Furthermore, we molecularly defined 22 blood group systems comprising 26 genes and 5 platelet antigen genes. Herein, 37% tested COVID-19 negative while 63% tested positive by PCR. Within the negative subjects, HLA-B*57:01, HLA-B*55:01, DRB1*13:01, and DRB1*01:01, were enriched, and in the positive group, homozygosity for DQA/DQB, DRB1*09:01, and DRB1*15:01 was observed. For HLA-DQA1, we observed an enrichment for DQA1*01:01, DQA1*02:01, and DQA1*01:03. For HLA-DQB1, we found that HLA-DQB1*06:02 was enriched in the positive group, while HLA-DQB1*05:01 and HLA-DQB1*06:03 were enriched in the negative group. The homozygous platelet antigen HPA-1a was significantly enriched in the negative group, contrasting with the HPA-1ab, which was enriched in the COVID-19 infected group.

**Abstract:**

Herein, we included 527 individuals from two Hospitals, Chemnitz and University-Hospital Leipzig. In total, 199 were negative for PCR and 328 were positive upon first admission. We used next generation sequencing for HLA-A, B, C, DRB1, DRB345, DQA1, DQB1, DPA1, and DPB1, and in some cases, HLA-E, F, G, and H. Furthermore, we molecularly defined 22 blood group systems comprising 26 genes and 5 platelet antigen genes. We observed a significant enrichment of homozygosity for DQA/DQB in the positive group. Within the negative subjects, HLA-B*57:01, HLA-B*55:01, DRB1*13:01, and DRB1*01:01 were enriched, and in the positive group, homozygosity for DQA/DQB, DRB1*09:01, and DRB1*15:01 was observed. DQA1*01:01, DQA1*02:01, and DQA1*01:03 were enriched in the negative group. HLA-DQB1*06:02 was enriched in the positive group, and HLA-DQB1*05:01 and HLA-DQB1*06:03 were enriched in the negative group. For the blood group systems MNS, RH, LE, FY, JK, YT, DO, and KN, enrichment was seen in both groups, depending on the antigen under observation. Homozygosity for D-positive RHD alleles, as well as the phenotypes M-N+ of the MNS blood group system and Yk(a-) of the KN system, were enriched in the positive group. All of these significances disappeared upon correction. Subjects who carried homozygous HPA-1a were more frequent in the negative group, contrasting with the finding that HPA-1ab was enriched in the positive group.

## 1. Introduction

The 2020–2022 COVID-19 (coronavirus) pandemic, caused by the SARS-CoV-2 virus, affected a large part of the population worldwide. This was especially true in Saxony, which showed the highest infections rates (46.8% in November 2022) in Germany [1]. Older people and individuals with previous illnesses, particularly, had reduced chances of survival. In daily life, pathogens induce immune responses in humans via two main pathways: the innate and the adaptive immune system, introduced by [2]. Herein, we concentrate on the later immune reactivity towards the SARS-CoV-2 virus [3]. The term immunogenetics describes a variety of genes involved in immunological responses against pathogens. These are inherited and generally polymorphic, i.e., have a great variability. The first Nobel Prize in the field of immunogenetics was awarded to Baruj Benacerraf, Jean Dausset, and George Davis Snell in 1980 for discovering genetically determined cellular surface structures, which control immunological reactions: “The Nobel Prize in Physiology or Medicine 1980” [4]. The genes encoded by the MHC (major histocompatibility complex) region, the HLAsystem (human leukocyte antigen), are expressed on the surface of nucleated cells and act as the counterpart of T and NK (natural killer) cell receptors, contributors to innate immunity. Upon infection, peptides of the intruder are presented at the cell membrane in the context of the MHC molecules: HLA-DR, DQ, and DP present peptides to the CD4 (cluster of differentiation) positive helper T cells, and HLA-A, B, and C, as well as the non-classical genes HLA-E and HLA-G, will present peptides to the CD8 positive cytotoxic T cells. HLA-F seems to present large peptides, but its precise function has not yet been elucidated [5]. The HLA nomenclature has been established through several workshops and accepted by the WHO nomenclature committee for factors of the HLA-System [6].

The MHC, as part of the immune system, plays a crucial role in the defense against pathogens [7]. Special variants of the MHC are associated with increased risk and complications of diseases, and have been found to be associated with viral infections [8,9]. For example, HIV is predominantly associated with MHC class I alleles, and Hepatitis B and Papilloma virus infections with MHC class II. In contrast, Dengue fever and Hepatitis C are associated with MHC class I and II alleles or allele groups [8]. An association between HLA-B27 and HLA-B57 was described for HIV [10], SARS-CoV [11], and SARS-CoV-2 [12]. In addition, a number of allelic variants of blood groups are also linked to infection susceptibility [13,14]. Herein, we searched for genetic factors within MHC and blood groups, which may determine susceptibility or resistance to SARS-CoV-2 infection. Genetic variants of the human SARS-CoV-2 receptors might be taken into consideration [15]. On the other hand, blood groups, which are, in general, multi-allelic and encoded on several regions of the human genome, were shown to be associated with viral infections [13]. We hypothesize that the previous observations of the propensity to develop immune recognition, and, hence, immune reaction, against SARS-CoV-2 depends on the allele status of HLA and blood group molecules in the infected individuals.

Several groups worldwide have reported associations with alleles/antigens of the HLA system. Besides single antigens, antigen groups or alleles were found to be associated with SARS-CoV-2 infection, severity, death, and survival [16,17,18,19,20,21,22,23]. However, the number of individuals tested was relatively small, and underlying diseases such as end-stage renal disease, autoimmunity, and immunosuppression might have affected the outcome. For the human platelet system (HPA), no data are yet available. For the common blood groups ABO and RHD, associations were observed, but the observations could not always be confirmed [14,17,18,24,25,26,27].

We analyzed, by next generation sequencing (NGS), the 11 classical loci for HLA (A, B, C, DRB1, DRB3, DRB4, DRB5, DQA1, DQB1, DPA1, and DPB1) and, for some of the individuals, the non-classical HLA genes, for a total of 26 red cell blood groups and 5 groups of the HPA system. Several of the alleles or allele groups were found to be primarily associated with the SARS-CoV-2 infection, with a *p* value of <0.1.

## 2. Materials and Methods

### 2.1. Cohort

The examined unrelated cohort includes 527 samples that were collected during the period from 26 May 2020 to 31 March 2022, in the federal state of Saxony, Germany. Of these, 328 were from subjects who tested positive for SARS-CoV-2 by PCR, and 199 samples were from individuals who tested negative for SARS-CoV-2 by PCR or who were not infected, as detected by antibody testing. The cohort included individuals ranging in age from 15 to 88 years at the time of sampling. Of these, 190 were male and 337 were female individuals.

### 2.2. DNA Preparation

DNA was isolated from EDTA blood samples according to the manufacturer’s recommendation (QIAamp DNA Blood Mini Kit, QIAGEN, Hilden, Germany).

### 2.3. Next-Generation Sequencing HLA Typing

HLA typing was performed using commercial test kits, the Alltype NGS 11-Loci (One Lambda, West Hills, CA, USA) and AlloSeq^®^ Tx 17 (CareDx, San Francisco, CA, USA). The sequencing was performed on a MiSeq Sequencing device (Illumina, San Diego, CA, USA), strictly following the manufacturer’s recommendations. The laboratory in Leipzig that performed the HLA NGS typing has a European accreditation for NGS HLA typing.

### 2.4. Next-Generation Sequencing Blood Group Typing (BG) and Platelet Antigen Genes (HPA)

The entire blood group typing workflow, including NGS library preparation and analysis software, is accredited to the American Society for Histocompatibility and Immunogenetics (ASHI).

The test system included 22 blood group systems, comprising 26 genes and 5 platelet antigen genes (Table 1). Enumeration of nucleotides and exons, as well as primer design, are based on the specified reference sequences for transcripts and genomic DNA.

#### 2.4.1. Regions of Interest

A four-primer-amplicon-based method was used. To achieve high-resolution results for genotyping, we chose individual amplification strategies for each of the BG and HPA systems. Most amplicons covered an entire exon, or, in the case of exons larger than 511 bp, a part of an exon. Exons smaller than 511 bp were adjoined by intron sequences of variable size. In addition, some amplicons target pseudoexons (GYPB, GYPE) or parts of the 5′-UTR (ACKR1) or the promoter (P1PK). Depending on the blood group system, the number of regions of interest differed between the analyzed BG genes (Appendix A).

#### 2.4.2. Primer Design

Primers were designed to yield amplicons 313 bp to 511 bp in size with the online tool Primer3Plus (https://www.primer3plus.com/index.html access date: 1 November 2022) (Appendix A). All primers were screened for additional SNPs using the SNPCheck software (https://genetools.org/SNPCheck/snpcheck.htm access date 1 November 2022). Where necessary, additional primers were included to avoid allele drop-outs. Each primer pair was checked for specificity using Sanger sequencing analysis. For typing the MNS blood group, we used generic primers to amplify GYPA, GYPB, and GYPE exons simultaneously. Similarly, generic primers were used for the amplification of RHCE and RHD exons, except for exon 10. Primers were purchased from Biolegio (Nijmegen, The Netherlands).

#### 2.4.3. Next-Generation Library Preparation

Amplicons were generated from genomic DNA in four multiplex PCR reactions. The reaction mixture for each sample consisted of 5 μL of KAPA2G Fast Multiplex Mix (Kapa Biosystems, Cape Town, South Africa), 1.5 to 4 pmol of each primer, and 20–50 ng of genomic DNA, for a total volume of 10 μL. The PCR protocol was carried out as follows: 95 °C—3 min; 30 cycles of 95 °C—15 s, 63 °C—1 min, 72 °C—2 min; 72 °C—3 min. After a clean-up step using para-magnetic beads, a second-round PCR served to add sample-specific barcodes and Illumina-specific adapter sequences. This barcoding-PCR was carried out using the following conditions: 3 μL purified PCR product, 7.5 μL KAPA HiFi HotStart ReadyMix (Kapa Biosystems, Cape Town, South Africa), and 3 μL barcoding primer mix (0.4 μM, each), for a total volume of 15 µL; 95 °C—5 min; 15 cycles 95 °C—20 s, 65 °C—15 s, 72 °C—20 s; 72 °C—1 min. PCR products, which were obtained for 190 different samples, were pooled and quantified using a Quantus Fluorometer (Promega GmbH, Walldorf, Germany). For sequencing, 7 pM of the NGS library was applied to a MiSeq instrument (Illumina Inc., San Diego, CA, USA) for a paired end run with 272 and 260 cycles for Read 1 and Read 2, respectively, using a standard v3 cartridge according to the manufacturer’s instructions. As an internal run control, we used a spike-in of 15% PhiX.

#### 2.4.4. Evaluation Criteria

Runs that were included in this study needed to have the following criteria: cluster density must be between 800 and 1300 k/mm^2^, more than 80% of clusters pass the filter, and Q30 score must be higher than 80%.

### 2.5. Analysis of the NGS Data

HLA: The HLA results were determined by analyzing the fastq files in the Type Stream Visual (TSV) software v2.0 (One Lambda) with the AlloSeq Assign v1.02 software (CareDx).

Blood grouping and HPA: These were performed using the accredited and self-developed analysis software Blood Group Analyzer (Institute for Transplantation Diagnostics and Cell Therapeutics, University Hospital Düsseldorf, Düsseldorf, Germany). Briefly, the following steps were applied to the raw data: The MiSeq reporter software was used to generate a pair of fastq files for each sample. The PANDAseq paired-end assembler [28] served to align the Illumina reads and reconstruct an overlapping sequence. After allocation of each reconstructed sequence to a specific region of interest using a decision tree, the amplification primer sequences were deleted from the sequence. To remove sequencing errors that were introduced during PCR amplification or by the MiSeq device, we performed an error correction step, taking all sequences which belong to the same region of interest into account. By comparing the error-corrected sequences with a table of reference sequences, which comprised all known alleles of the specific region, allele groups were generated. The final allele determination was performed by forming intersections of the allele groups for all exons for a locus.

Genotyping results were accepted if the entire set of fragments for a locus were each present, with the appropriate depth of coverage of 80 for homozygous fragments and 30 from each allele group for heterozygous fragments.

#### 2.5.1. Allele Assignment and Phenotype Determination (BG and HPA)

##### Referenced Alleles

Allele names and phenotypes were assigned according to the nomenclature described by the ISBT working parties on 2 November 2022 (https://www.isbtweb.org/isbt-working-parties/rcibgt/blood-group-allele-tables.html, accessed on 15 November 2022).

##### Non-Referenced Alleles

For non-referenced alleles, a provisional nomenclature was chosen that unambiguously identified additional amino acid and nucleotide changes (Appendix A). We abandoned the assumption of an RBC phenotype for these alleles, because uncharacterized mutations might alter the antigenicity of the background allele.

### 2.6. Statistical Analysis

For individual analysis of the HLA alleles and blood groups, we applied a t-statistic of positive PCR proportion of the respective subcohort against the overall positive rate. The resulting *p*-values were corrected for multiple testing using the Bonferroni method. Alleles and blood groups with a corrected *p*-value < 0.05 were selected for subsequent enrichment analyses of persons with positive or negative PCR using Fisher’ exact test with subsequent Bonferroni correction.

Combinations of two to four HLA alleles and/or blood groups were eventually evaluated, in terms of corresponding positive PCR proportion and enrichment analyses, by Fisher’s exact test with Bonferroni correction.

## 3. Results

### 3.1. Review of Current Results on COVID-19 Infections

Table 2 summarizes some of the results found in the literature. Obviously, depending on the focus of the given patient group, different associations were observed. Furthermore, region, nationality, and numbers of probands differed from observation to observation. In our study, we distinguished two subgroups: COVID-19-infected people and non-infected individuals. We analyzed the frequencies of the investigated genetic markers (HLA, blood groups, HPA) of the Saxon population and compared the findings in the COVID-19-infected and non-infected groups with the total Saxonian population.

### 3.2. Analysis of the HLA Alleles

Data on HLA class I and II typing is available for 527 persons. Class I data comprises 33 alleles of HLA-A, 50 alleles of HLA-B, and 30 alleles of HLA-C. For class II, we obtained 34 alleles of DRB1, 13 alleles of DRB345, 17 alleles of DQA1, 19 alleles of DQB1, 7 alleles of DPA1, and 26 alleles of DPB1. The overall HLA allele frequencies are depicted in Appendix A. In addition to the classical HLA genes, the frequencies of the non-classical HLA genes E, F, G, H, MICA, and MIC-B are also presented in Appendix A. These were obtained from a subsample of 134 individuals in the current cohort.

#### 3.2.1. Homozygosity of HLA Alleles

Within the numerous reports of possible HLA associations with different stages of the SARS-CoV-2 infection and disease, the degree of homozygosity was found to be a parameter influencing the course of COVID-19 infection [21,22]. We analyzed this parameter in our cohort and were able to support this idea (Figure 1). In particular, we observed an association between PCR status and homozygosity of the loci DQA1 and DQB1, with *p*-values of 0.01 and 0.04, respectively. These two molecules form the HLA-DQ molecule. All the other genes showed no significant deviation for homozygosity between the PCR positive and PCR negative individuals in the study.

#### 3.2.2. Association of HLA Alleles in the Study

We evaluated associations of the individual alleles of each HLA gene with the infection status of the corresponding persons (Figure 2a–d): For each gene, the fractions of persons with positive PCR tests were calculated for the different alleles and compared to the overall PCR positive rate of 62.2%. In total, 328 persons were contained in the positive subcohort; 199 in the negative.

We found alleles with PCR positive rates ranging from 100% to 0%; however, more frequent alleles with n > 10 showed rates between approximately 50% and 70%. Application of t-statistics revealed that no alleles of the HLA, A, or C genes significantly differed from the overall rate (Figure 2b,d). Contrary, two alleles of the B locus, B*57:01 and B*55:01, showed corrected *p*-values <0.001 and <0.05, respectively. They showed decreased infection rates of about 40%, and were selected for downstream enrichment analyses, as discussed below.

Analyses of HLA class II alleles were performed analogously to class I alleles. In the first step, alleles which were associated with an increased or decreased PCR positive rate were identified (Figure 3a; corresponding t-statistics are given in Appendix A). Comparison of allele-wise PCR rate with the overall rate of 62% revealed an association of alleles DRB1*09:01 and DRB1*15:01 with the positive subcohort, and an association of DRB1*13:01, DRB1*01:01, and DRB1*13:05 with the negative, not-infected subcohort. For HLA-DQA1, we observed decreased PCR positive rates for DQA1*01:01, DQA1*02:01, and DQA1*01:03. For HLA-DQB1, we found that HLA-DQB1*06:02 was associated with the positive, and HLA-DQB1*05:01 and HLA-DQB1*06:03 with the negative subcohort. For the other HLA genes, no associations were observed, with the exception of DPB1*20:01.

We then collected all class I and II alleles with corrected *p*-values <0.05 in a subsequent enrichment analysis (Figure 3b). Therefore, we tested, individually for each HLA locus, whether individuals who carry a particular allele were more common in the PCR positive or negative subcohorts. In general, we found that HLA class I and II alleles were enriched in both subcohorts; however, 11 of the 14 selected alleles were enriched in the negative subcohort. This indicates that HLA characteristics tend to act protectively with regard to COVID-19 infection, rather than making a person susceptible.

Ten alleles were found to be enriched in positive or negative persons, with a *p*-value <0.05. However, two of them (B*55:01 and DPB1*20:01) were found in fewer than 20 individuals, limiting the reliability of this analysis. For the other alleles, the strongest enrichment was found for B*57:01 (*p*_corrected_ = 0.01), B*55:01 (*p*_corrected_ = 0.08, but *n* = 14 only), and DQB1*06:02 (*p*_corrected_ = 0.06).

The non-classical HLA antigens were not very polymorphic in the cohort studied, such as HLA class I, as shown in Appendix A. An association with COVID-19-infected or non-infected individuals is, therefore, not to be expected, even considering the relatively small number of people tested (*n* = 134).

### 3.3. Enrichment of Blood Group and Platelet Antigens in the Two Subgroups

To define possible factors associated with the infection of SARS-CoV-2 in our population, we analyzed blood group and platelet antigens by NGS. Since no molecular blood group typing is usually prescribed in Germany, there are little data available. To overcome this gap, herein, we present the blood group data of 521 persons in our cohort.

In total, up to 48 alleles in 31 blood group systems are available. The overall frequencies of these characteristics in our study population are given in Appendix A.

We applied t-statistics to select blood groups with PCR rates differing from the overall rate of 62% positive. In total, we identified 25 blood groups in 12 systems with a corrected *p*-value < 0.05 (Figure 4a; corresponding t-statistic plots and plots of the remaining 19 systems are shown in the Appendix A).

In total, four platelet antigen groups showed differential PCR positivity: HPA-15ab and bb showed increased and decreased positive rates, respectively, which could not be confirmed in enrichment analyses (Figure 4b). In contrast, HPA-1 showed strong enrichment of positive (HPA-1ab; *p*_corrected_ = 0.03), negative, and non-infected persons (HPA-1aa; *p*_corrected_ = 0.03). In addition, we observed differential PCR rates for the blood groups ABO, MNS, P1PK, RH, and LU, as shown in Figure 4b. For the common blood group system ABO, no significant enrichment could be found, in contrast to previous studies [14,17,29]. Other blood group systems, such as MNS, Rhesus, Lewis, Duffy, Kidd, Carthwright, Dombrock, and Knops showed enrichment in one of the groups of the investigated cohort (Figure 4b). Therefore, enrichment of RH variant DD in the positive and of MNS M-N+ in the negative subcohort stands out, but the corrected *p*-values exceed 0.1. The findings of Matzhold EM. Et al. 2021 [25], describing Lewis a-b- as a protective factor, could not be confirmed in our study population (*p*-values > 0.1; Figure 4b).

### 3.4. Correlation of HLA, Blood Group and Platelet Antigens

After individual evaluation of single HLA alleles and blood groups, we aimed to conduct integrated enrichment analyses of the two immunogenetic realms. Therefore, subcohorts of individuals were generated, and showed combinations of two, three, and four characteristics identified in the previous subchapters. For association with the PCR positive subcohort, we selected two HLA alleles with *p*-value < 0.05 (DRB1*15:01 and DQB1*06:02) and three blood groups (HPA-1ab, M-N+, and RH DD). To increase the coverage of this analysis, we extended the *p*-value threshold for blood groups to 0.1 to select more characteristics.

Each single characteristic is related to a PCR positive rate of about 70%, which is 8% higher than the overall rate (Figure 5a; diagonal of the matrix). When combined with another characteristic, this proportion increases markedly (Figure 5a; lower triangle). Combinations of DRB1*15:01 or DQB1*06:02 with either HPA-1ab or MNS M-N+, especially, show a PCR positive rate of 83%, which is 20% higher than the average. The corresponding *p*-values are below 0.05; however, they are based on only 23–36 persons (Figure 5a; upper triangle). Combinations of three and four of the characteristics were also evaluated (Figure 5b), and mainly involved DRB1*15:01 and DQB1*06:02 alleles due to the large overlap of the corresponding subcohorts. In brief, we found 36 persons with DRB1*15:01 and DQB1*06:02 alleles as well as HPA-1ab blood group. Of these, 30 (=83%) were grouped into the PCR positive cohort, which leads to a corrected *p*-value of 0.04. These numbers are almost identical to the pairwise combinations of DRB1*15:01 and HPA-1ab and DQB1*06:02 and HPA-1ab.

Regarding potentially protective characteristics, eight HLA alleles (B*57:01, B*55:01, DRB1*13:01, DRB1*01:01, DQA1*01:01, DQA1*01:03, DQB1*05:01, and DPB1*20:01; enrichment *p*-values < 0.05) and one blood group (HPA-1aa; *p*-value = 0.03) were included in combinatorial analysis (Figure 5c). A total of 37 persons with B*57:01 showed a PCR positive rate of 41%, 19% below average. When combined with HPA-1aa, this proportion dropped to 35%, with the most significant enrichment of PCR negative persons (*p*_corrected_ = 0.04). All other investigated combinations of HLA alleles and HPA-1aa showed proportions below 50%, which was at least 10% below the 60% PCR positivity in the HPA-1aa subcohort without combinatorics. Of these, DQA1*01:01 and DQB1*05:01 combined with HPA-1aa showed the most enrichment.

Besides the combination B*57:01 & HPA-1aa, the combination of DQA1*01:01, DQB1*05:01, and HPA-1aa revealed strong enrichment (*p*_corrected_ = 0.035) and subcohort PCR positivity of 49% (Figure 5d). Other potentially protective combinations are the following: DRB1*13:01, DQA1*01:03, and HPA-1aa (*p*_corrected_ = 0.04); DRB1*01:01, DQA1*01:01, DQB1*05:01, and HPA-1aa (*p*_corrected_ = 0.07); and DRB1*01:01, DQA1*01:01, and DQB1*05:01 (without HPA-1aa; *p*_corrected_ = 0.04), all of these showing PCR positive rates of approximately 50%.

## 4. Discussion

One of the lessons of such a study as the one presented herein, and of all associative studies, is that the SARS-CoV-2 infection, course of COVID-19 disease, and, in some cases, death of the patients, constitute a multifactorial and multigenetic problem. The hope that singular associations could be identified has not come true. Different studies have included patients from different populations and with additional comorbidities, such as kidney transplant recipients, autoimmune patients, or others. The partially different HLA associations with COVID-19 infection in different populations (see Table 2) [29] are not necessarily contradictory. Rather, this shows a diverse escape mechanism of the virus in different populations. Herein, the high-frequency HLA alleles/haplotypes were shown to be different, and have different binding affinities to the SARS-CoV-2 virus [30,31,32,33,34,35,36].

We observed an influence of homozygosity of the loci DQA1 and DQB1 in the COVID-19 infected group. Homozygosity is thought to be associated with lower immunocompetency, resulting in rapid disease progression after virus infection [37]. The products of these genes form a heterodimer on the cell surface. A lower SARS-CoV-2 peptide load in the MHC may promote the disease. Similar data have also been found for other diseases, including autoimmune ones. In addition, we found that specific HLA class I alleles, e.g., HLA-B*57:01 and HLA-B*55:01, were enriched in the negative PCR group, indicating that they might present crucial peptides to the immune cells. The first, HLA-B*57:01, remained significant after Bonferroni correction as the only allele in the present study. This allele plays a pivotal role in HIV infection [38,39,40,41,42,43,44,45,46]. It might well be that similar mechanisms play a role here. One should keep in mind that MHC class I alleles are the counterpart of cytotoxic T cell receptors, which are used to eliminate infected cells. Additional studies are needed to elucidate this problem. In addition to the class I alleles, several class II alleles were found to be significant, but failed to gain significance after Bonferroni correction. This is also the case for most of the studies reported so far, wherein the number of individuals tested was low. Therefore, meta-analyses are needed. Interestingly, the uncorrected results of the HLA-DR/DQ alleles point to an additional influence of the MHC and the immune system triggered by these proteins. These alleles HLA-DRB1*15:01 and DQB1*06:02 usually form a haplotype. Both were seen in the positive group in contrast to HLA-DRB1*13:01 and DQB1*05:01, which were enriched in the negative group, as were HLA-DRB1*01:01 and HLA-DQB1*06:03. Whether or not these associations are due to one of the alleles and the other is an innocent bystander will be a focus for further analyses. It might also be that the haplotype is important for the disease, per se. Nevertheless, class II proteins induce and regulate the immune response, and are needed for triggering this towards the virus. The molecule HLA-B*57:01 is protective with respect to SARS-CoV-2. Individuals carrying this allele are less prone to becoming infected; at least, this was the case for the first two waves [47]. Similar data with respect to HIV and SIV have been reported earlier. Homozygosity is associated with reduced immune competence, which leads to a more rapid progression of viral diseases. MHC heterozygosity results in more alleles leading to immune responses against more pathogens, potentially increasing successful defense and survival rates against a wider spectrum of infectious diseases. We were not able to find any association with HLA-A, HLA-C, HLA-DPA1, and HLA-DPB1.

Among all common and uncommon blood group systems and platelet antigens, we found 12 systems with significant differences when the PCR positive and negative cohort were analyzed. For the common blood group system ABO, no significant association could be found in contrast to previous studies [14,17,24]. For some antigens of other blood group systems, such as MNS, RH, LE, FY, JK, YT, DO, and KN, we found enrichment in one of the groups of the investigated cohort (Appendix A). Homozygosity for D-positive RHD alleles, as well as the phenotypes M-N+ of the MNS blood group system and Yk(a-) of the KN system, were enriched in the PCR positive group. The findings of Matzhold et al. 2021 [25], which regarded Le(a-b-) as a protective factor, could not be confirmed in our study. All of these significances disappeared upon correction.

In the early days of the pandemic, a genome-wide analysis on the association of a viral infection with specific blood groups was conducted. We, however, could not confirm these results. On the other hand, several blood group antigens showed a significant enrichment in either group, which did not reach significance upon correction. Therefore, these findings could be regarded as a trend, which might reach significance if an increased number of subjects had been tested. The association of the glycoprotein GPIIIa, which is part of the heterodimeric integrin receptor GPIIb/IIIa, was expected if one considers that platelets can bind the virus [44]. In addition, it has been shown that activation of the receptor is downregulated in COVID-19 infected platelets. The degree of GPIIb/GPIIIa dysfunction was correlated with disease severity [48]. Here, we show an association of SARS-CoV-2 infection with a single nucleotide polymorphism in *GPIIIa*. A substitution of cytosine for thymidine at position 1565 in exon 2 of the *GPIIIa* gene leads to an amino acid exchange at position 33: a leucine (HPA-1a) or a proline (HPA-1b) [49]. The Pro^33^ isoform was shown to influence platelet shape change, secretion, and clustering of integrins [50]. This isoform might enhance the binding capacities of SARS-CoV-2 and facilitate virus uptake.

Despite the limitations of our study, the data presented herein clearly show that COVID-19 infection and all of its consequences are multifactorial and multi-genetic. We were able to define HLA alleles and a member of the HPA-system as being associated with the infection in the beginning of the pandemic. In this study, we did not focus on the severity of COVID-19 infection with different symptoms, nor did we evaluate previous illnesses of the individuals studied. A possible role of other immunologically relevant molecules, e.g., cytokines, in SARS-CoV-2-infected patients with “cytokine storm” was described by Coperchini et al. 2020 [51], Chi et al. 2020 [52], Huang et al. 2020 [53], and Merad and Martin 2020 [54]. Further comprehensive studies of cytokines and cytokine polymorphism may lead to a further understanding of COVID-19 and its complications. Establishing associations of COVID-19 with the polymorphic HLA system and its molecules, as well as variants of the virus, may reveal the possibility to direct treat high-risk populations, e.g., immunocompromised patients. Usually, these NGS data are available in the respective institutions. However, one should take into consideration that the virus is undergoing continuous evolutionary mutation and selection, leading to possible immune-evasive viral variants. Therefore, associations represent momentum in science.

## Figures and Tables

**Figure 1 biology-12-00037-f001:**
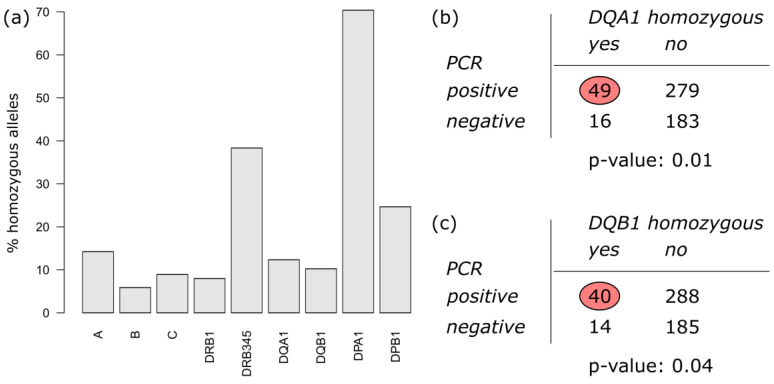
Distribution of HLA allele homozygosity in the cohort: (**a**) percent of homozygous alleles among the different genes. (**b**,**c**) 2 × 2 contingency tables of DQA1 and DQB1 homozygosity, stratified by PCR status, are shown along with corresponding *p*-values obtained from Fisher’s exact test.

**Figure 2 biology-12-00037-f002:**
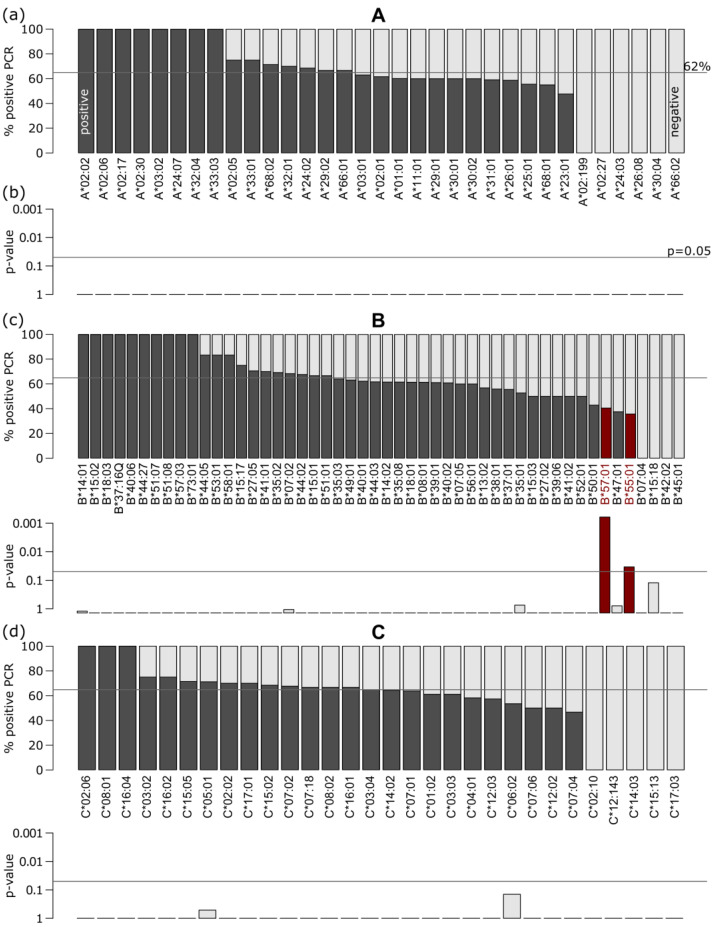
Analysis of the individual class I alleles in the PCR positive/negative subcohort: (**a**) Percent of persons with positive PCR stratified by all alleles observed for HLA-**A**. Black and gray bars show the proportion of PCR positive and negative persons for a particular allele, respectively. The overall positive PCR rate of 62% is indicated by the horizontal line. (**b**) t-statistics of positive PCR proportion against the overall positive rate provided *p*-values, which were corrected for multiple testing using the Bonferroni method. The horizontal line indicates a corrected *p*-value of 0.05. (**c**) Positive PCR proportions and corresponding *p*-values of the HLA-**B** locus alleles. Alleles with corrected *p*-value < 0.05 are highlighted in red. (**d**) HLA-**C** Locus alleles; see above.

**Figure 3 biology-12-00037-f003:**
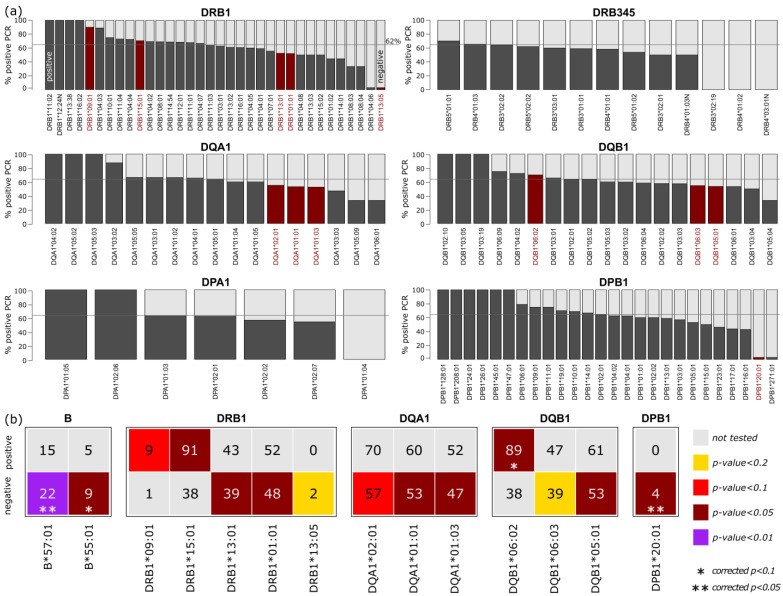
Analysis of the class II alleles with regard to PCR status: (**a**) Percent of persons with positive PCR stratified by the alleles of DRB1, DRB345, DQA1, DQB1, DPA1, and DPB1 loci. Alleles with corrected *p*-values < 0.05 are highlighted in red (see description of Figure 2). T-statistics of the loci are given as Appendix A. (**b**) Contingency tables of the selected HLA class I and II alleles (corrected *p*-value < 0.05). Numbers refer to all persons with the respective allele, split by positive and negative PCR status. Colors indicate significance of enrichment as obtained from Fisher’s exact test (see figure legend). Enrichments with *p*-values < 0.1/0.05 after Bonferroni correction are highlighted with asterisks.

**Figure 4 biology-12-00037-f004:**
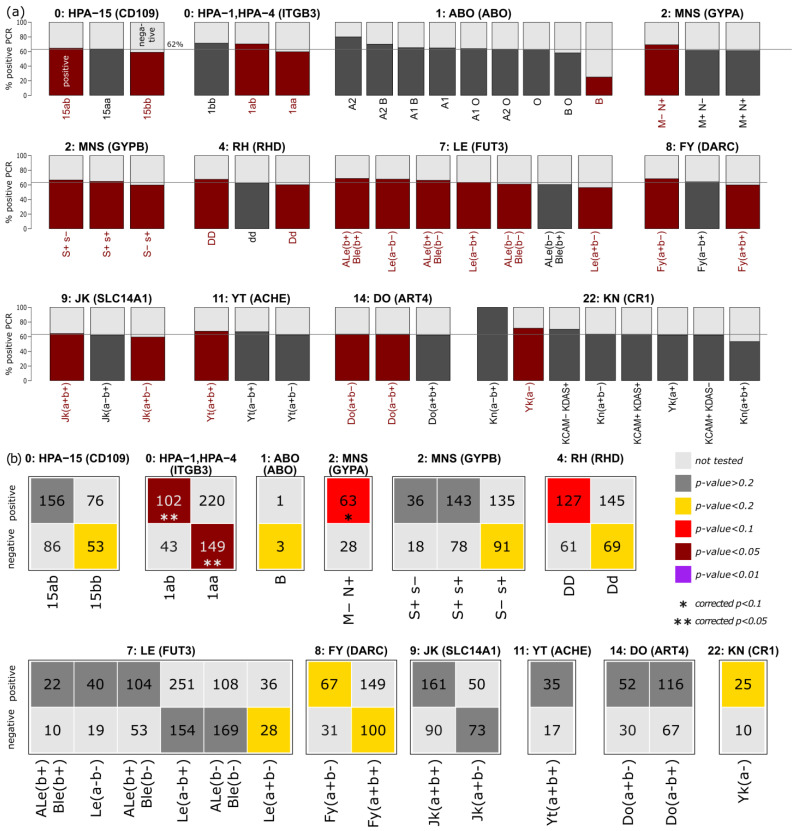
(**a**) Percent of persons with positive PCR, stratified by the blood groups. Groups with corrected *p*-values < 0.05 are highlighted in red (see description of Figure 2). (**b**) Contingency tables of the selected phenotypes (corrected *p*-value < 0.05). Numbers refer to all persons with the respective phenotype, split by positive and negative PCR status. Colors indicate significance of enrichment as obtained from Fisher’s exact test (see figure legend). Enrichments with *p*-values < 0.1/0.05 after Bonferroni correction are highlighted with asterisks.

**Figure 5 biology-12-00037-f005:**
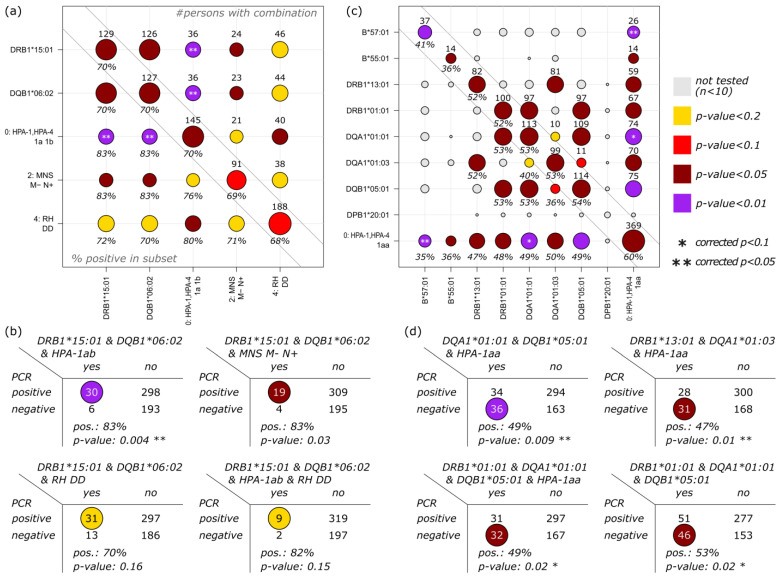
Combined analysis of HLA alleles and blood groups: (**a**) pairwise combinations of alleles and/or blood groups identified to enrich PCR positive persons are shown, together with the number of persons with the particular characteristics (upper triangle) and proportion of PCR positive persons within this subcohort (lower triangle). Colors indicate enrichment, as obtained from Fisher’s exact test. Enrichments with *p*-values < 0.1/0.05 after Bonferroni correction are highlighted with asterisks. Numbers and percentages along the diagonal reflect properties of single HLA alleles or blood group characteristics, as discussed above. (**b**) Combinations of three or four of the selected characteristics were evaluated for enrichment in the positive subcohort in terms of 2 × 2 contingency tables of each phenotype. *p*-values were calculated by Fisher’s exact test; see color legend. Combinations with fewer than 10 persons were rejected from this analysis. (**c**) Pairwise combinations of alleles and/or blood groups enriched in the negative subcohort. (**d**) Enrichment of combinations of three or four characteristics in the negative subcohort. Asteriks indicates the corrected *p*-values.

**Table 1 biology-12-00037-t001:** Blood group systems and HPA included in the NGS test system.

ISBT System No.	System Name	System Symbol	Gene(s)	Reference Sequence (Transcript)	Reference Sequence (Genomic)
001	ABO	ABO	*ABO*	NM_020469	NG_006669.2
002	MNS	MNS	*GYPA*	NM_002099.6	NG_007470.3
			*GYPB*	NM_002100.4	NG_007483.2
			*GYPE*	NM_002102.3	NG_009173.1
003	P1PK	P1PK	*A4GALT*	NM_017436.4	NG_007495.1
004	Rhesus	RH	*RHCE*	NM_020485.4	NG_009208.3
			*RHD*	NM_016124.3	NG_007494.1
005	Lutheran	LU	*B-CAM*	NM_005581.4	NG_007480.1
006	Kell	KEL	*KEL*	NM_000420.2	NG_007492.3
007	Lewis	LE	*FUT3*	NM_000149.4	NG_007482.2
008	Duffy	FY	*DARC*	NM_002036.3	NG_011626.2
009	Kidd	JK	*SLC14A1*	NM_015865.6	NG_011775.4
010	Diego	DI	*SLC4A1*	NM_000342.3	NG_007498.1
011	Cartwright	YT	*ACHE*	NM_015831.2	NG_007474.2
014	Dombrock	DO	*ART4*	NM_021071.2	NG_007477.2
015	Colton	CO	*AQP1*	NM_198098.2	NG_007475.2
018	H	H	*FUT1*	NM_000148.3	NG_007510.1
			*FUT2*	NM_000511.5	NG_007511.1
020	Gerbich	GE	*GYPC*	NM_002101.4	NG_007479.1
021	Cromer	CROM	*CD55*	NM_000574.3	NG_007465.1
022	Knops	KN	*CR1*	NM_000573.3	NG_007481.1
023	Indian	IN	*CD44*	NM_0010011391.1	NG_008937.1
032	Junior	JR	*ABCG2*	NM_004827.2	NG_032067.2
033	Langereis	LAN	*ABCB6*	NM_005689.2	NG_032110.1
034	Vel	VEL	*SMIM1*	NM_001163724.3	NG_033869.1
036	Augustine	AUG	*SLC29A1*	NM_004955.2	NG_042893.1
	HPA-1/HPA-4		*ITGB3*	NM_000212.2	NG_008332.2
	HPA-2		*GP1BA*	NM_000173.7	NG_008767.2
	HPA-3		*ITGA2B*	NM_000419.5	NG_008331.1
	HPA-5		*ITGA2*	NM_002203.4	NG_008330.2
	HPA-15		*CD109*	NM_133493.3	NG_033971.1

**Table 2 biology-12-00037-t002:** Overview of some publications in the context of HLA and COVID-19 association.

Population	Methods	Cohort	HLA Association Results	Reference
Russian	NGS, BAP	111 deceased patients with COVID-19	A*01:01 homozygosity high risk A*02:01, A*03:01 protective	[22]
British	NGS	147 European COVID-19 infected patients with variable outcomes (49 severe, 69 asymptomatic hospital staff positive for COVID-19)	DRB1*04:01 protective DQA1*01:01-DQB1*05:01-DRB1*01:01 higher frequency in asymptomatic group	[20]
Italian/Spanish	GWAS	835 (Italy) and 775 (Spain) COVID-19 infected patients from 7 hospitals	no significant HLA association chromosome 3p21.31, the peak association signal covered a cluster of six genes (SLC6A20, LZTFL1, CCR9, FYCO1, CXCR6, and XCR1) higher risk blood group A protective blood group O	[17]
Sardinian	SSP-PCR	182 COVID-19 infected patients asymptomatic, pauci-symptomatic, hospitalized	A*02:05, B*58:01, C*07:01, DRB1*03:01 protective A*30:02, B*14:02, C*08:02, DRB1*08:01 higher risk	[21]
Spanish	SSO, BAP	45 COVID-19 patients with mild, moderate, and severe infection (only HLA class I)	SARS-CoV-2 binding capacity of different HLA alleles A2, C1 subtypes; homozygosity of HLA alleles	[19]
Chinese	NGS	82 COVID-19 infected patients	C*08:01G, B*15:27, B*40:06, DRB1*04:06, and DPB1*36:01 higher in COVID-19 patients DRB1*12:02, DPB1*04:01 lower in COVID-19 patients	[23]
Multi ethnic	BAP		A*02:02, B*15:03, C*12:03 highest predicted capacity for SARS-CoV-2 epitope presentation A*25:01, B*46:01, C*01:02 lowest predicted capacity for SARS-CoV-2 epitope presentation	[12]
United Arab Emirates Abu Dhabi	NGS	115 patients with mild, moderate, and severe SARS-CoV-2 infection age mean mild: 34 ± 14 age mean moderate/severe: 58 ± 15	A*03:01, B44, DRB1*15:01 heterozygosity significant association with severity A*26:01, B*51:01 negative association (protectiv)	[16]

Red: higher frequencies in COVID-19 patients than in control group; green: higher frequencies in controls. NGS: Next generation sequencing, BAP: in silico binding affinity prediction, SSO: sequence-specific oligonucleotide, GWAS: genome-wide association study.

## Data Availability

The data supporting this study’s findings are available from the corresponding author upon reasonable request. Data is additionally deposited at the LeipzigHealthAtlas and can be accessed upon request. A link will be generated upon manuscript acceptance.

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
