# Peer review of "Immunogenetic Predisposition to SARS-CoV-2 Infection"

_biology, 2022, doi:10.3390/biology12010037_

Round 1
Reviewer 1 Report
This paper will be of interest to the scientists who perform immunogenetics with COVID-19. In this manuscript, the author uses a next-generation sequencing approach analysis the gene data from 527 individuals, some are PCR positive, and some are negative. They study from the HLA group, red cell blood groups, and HPA groups, and finally, they observed that several alleles were primarily associated with SARS-CoV-2 infection. However, some minor issues must be addressed before the manuscript is acceptable for publication.
Page 2_ Paragraph (P) 1, The author should provide the full name of the abbreviation the first time when the acronym appears in the article. MHC, HLA, NK, DQ, DP, CD4, DRB, DQA, DQB, DPA, DPB…
Page 2_P2, replace “SARS-CoV-1” with “SARS-CoV-2”.
Page 8_Figure 1, The author should add a title to figure 1.
Page 11_ Figure 3: Please add a title to figure 3.
Author Response
Thanks to the reviewer 1 for the comments.
Page 2_P1: We provide the full name of abbreviation MHC, HLA, CD, HPA the first time when the acronym appears in the article (chapter "introduction"). HLA antigens such as DQ DP DRB and so on are proper names for which there is no full name version. Here we added the reference to the WHO nomenclature in the text.
Page 2_P2: we have corrected it
Page 8_Figure 1: we add a title to figure 1
Page 11_Figure 3: we add a title to figure 3

Reviewer 2 Report
The authors showed huge efforts gathering all the data. The presented data are huge. It raised the question, what clincal impact has all these data? Should we do a next sequence generation evaluation before we vaccinate the population, iin order to identify the high risk population. The auhtors mentioned only SARS CoV 2 positive. It this a unique covid population or different variants like alpha, Beta, Gamma , omicron. This would be helpful for better classifying the statement of the study
Author Response
Thank you or the very important remark. Since we do not possess viral sequences we can only correlate the PCR date to the official infection waves of the different COVID 19 mutants, ISSN 2569-5266, 2021
We mentioned in the discussion part a possible clinical impact for immunocompromised patients. However, the results should be reproduced by others before far-reaching decisions are made.
